# Multi-Omics Analysis in Mouse Primary Cortical Neurons Reveals Complex Positive and Negative Biological Interactions Between Constituent Compounds of *Centella asiatica*

**DOI:** 10.3390/ph18010019

**Published:** 2024-12-27

**Authors:** Steven R. Chamberlin, Jonathan A. Zweig, Cody J. Neff, Luke Marney, Jaewoo Choi, Liping Yang, Claudia S. Maier, Amala Soumyanath, Shannon McWeeney, Nora E. Gray

**Affiliations:** 1Department of Medical Informatics and Clinical Epidemiology, Oregon Health & Science University, Portland, OR 97239, USA; mcweeney@ohsu.edu; 2Department of Neurology, Oregon Health & Science University, Portland, OR 97239, USAsoumyana@ohsu.edu (A.S.); grayn@ohsu.edu (N.E.G.); 3BENFRA Botanical Dietary Supplements Research Center, Oregon Health & Science University, Portland, OR 97239, USA; marneyl@oregonstate.edu (L.M.); claudia.maier@oregonstate.edu (C.S.M.); 4Department of Chemistry, Oregon State University, Corvallis, OR 97331, USA; 5Knight Cancer Institute, Oregon Health & Science University, Portland, OR 97239, USA

**Keywords:** *Centella asiatica*, triterpenes, caffeoylquinic acids, mouse primary cortical neurons, transcriptome, metabolome, co-expression, omics integration

## Abstract

**Background:** A water extract of the Ayurvedic plant *Centella asiatica* (L.) Urban, family Apiaceae (CAW), improves cognitive function in mouse models of aging and Alzheimer’s disease and affects dendritic arborization, mitochondrial activity, and oxidative stress in mouse primary neurons. Triterpenes (TT) and caffeoylquinic acids (CQA) are constituents associated with these bioactivities of CAW, although little is known about how interactions between these compounds contribute to the plant’s therapeutic benefit. **Methods:** Mouse primary cortical neurons were treated with CAW or equivalent concentrations of four TT combined, eight CQA combined, or these twelve compounds combined (TTCQA). Treatment effects on the cell transcriptome (18,491 genes) and metabolome (192 metabolites) relative to vehicle control were evaluated using RNAseq and metabolomic analyses, respectively. **Results:** Extensive differentially expressed genes (DEGs) were seen with all treatments, as well as evidence of interactions between compounds. Notably, many DEGs seen with TT treatment were not observed in the TTCQA condition, possibly suggesting CQA reduced the effects of TT. Moreover, additional gene activity seen with CAW as compared to TTCQA indicates the presence of additional compounds in CAW that further modulate TTCQA interactions. Weighted Gene Correlation Network Analysis (WGCNA) identified 4 gene co-expression modules altered by treatments that were associated with extracellular matrix organization, fatty acid metabolism, cellular response to stress and stimuli, and immune function. Compound interaction patterns were seen at the eigengene level in these modules. Interestingly, in metabolomics analysis, the TTCQA treatment saw the highest number of changes in individual metabolites (20), followed by CQA (15), then TT (8), and finally CAW (3). WGCNA analysis found two metabolomics modules with significant eigenmetabolite differences for TT and CQA and possible compound interactions at this level. **Conclusions:** Four gene expression modules and two metabolite modules were altered by the four treatment types applied. This methodology demonstrated the existence of both negative and positive interactions between TT, CQA, and additional compounds found in CAW on the transcriptome and metabolome of mouse primary cortical neurons.

## 1. Introduction

Traditional and complementary medicines are often sought to improve or maintain health, and it is estimated that 50% of the population in industrialized countries use these approaches [1]. In the elderly, these interventions are primarily used to promote resilience [2] since many of these are botanical extracts with purported anti-aging effects [3,4,5,6]. However, claims are controversial, and few clinical trials have been conducted with these treatments [7].

The study of botanical extracts for healthcare is challenging, in part due to the complex mix of compounds found in these plants. Unlike single-chemical drugs or isolated natural products, botanical extracts may contain hundreds of compounds [8]. Many traditional practitioners believe that these botanical mixtures are more clinically effective than isolated compounds due to interactions between the constituents. Interactions can occur between individual compounds, groups or fractions of compounds, constituents from different parts of the plant, or even between compounds derived from different botanical species, as seen in complex traditional herbal formulas [9,10]. These mixtures can interact with biological systems in a way that is additive, synergistic, or antagonistic [9,11,12,13,14,15]. Fraction-based methods have been developed to evaluate the synergy between compounds in whole plant extracts using a combination of mass spectrometry, isolation of natural products, and synergy assays [16]. In a whole organism, the nature of these interactions can vary depending on the compounds and be considered pharmacokinetic or pharmacodynamic. Pharmacokinetic interactions affect the bioavailability of a constituent in an organism through changes in its metabolism or cellular transport induced by another constituent. Pharmacodynamic interactions occur when two compounds affect each other’s binding to targets or have similar or opposing biological activity mediated through interactions at different points in molecular pathways [11,13].

While synergy studies have traditionally utilized targeted assays, leveraging untargeted omic analysis methods [14,15] in pre-clinical models offers an effective way to understand multiple bioactivities and mechanisms of action that are modulated by this mix of compounds, including understanding the interactions between the compounds. For the current project, we propose to evaluate two untargeted molecular domains, transcriptomics and metabolomics, in an in vitro model system to broadly study compound interactions in the biological activity of compounds within a water extract (CAW) of the botanical *Centella asiatica* (L.) Urb., family Apiaceae. This plant is native to both Asia and Australia and has a history of use in several systems of traditional medicine, including traditional Chinese medicine and Ayurvedic medicine [17], for the treatment of various conditions, including cognitive impairment [18]. Current in vitro, in vivo, and clinical research also supports the neurological benefits of *Centella asiatica* [19,20,21,22,23,24,25,26,27,28,29,30]. CAW can evoke particularly potent neuroprotective effects both in vitro and in vivo that may be mediated by effects on dendritic arborization [19,30], increased synaptic density [19,24,30], mitochondrial biogenesis [19,23,30], and activation of endogenous antioxidant mechanisms [19,23,30].

Two classes of compounds have been associated with the health-promoting bioactivity of CAW: pentacyclic triterpenes (TT) and mono- and di-caffeoylquinic acids (CQA) [31], although the contribution of each compound group to the overall effects of CAW and the mechanisms by which the compounds can elicit those effects have not been fully elucidated. The goal of this project is to begin to determine the contributions to bioactivity of these two compound classes separately or combined, as well as in the complete CAW extract.

In this study, mouse primary cortical neurons were treated with either CAW, groups of TT or CQA compounds separately, or a combination of the two (TTCQA). The in vitro CAW concentration chosen (50 µg/mL) was previously found to increase dendritic arborization, synaptic density, mitochondrial biogenesis, and activation of antioxidant response in mouse primary hippocampal neurons [30,32,33]. In vivo, CAW administered in drinking water has also been shown to affect these same pathways in both the hippocampus and cortex of treated mice [24,29,34]. All compounds were applied at concentrations equivalent to their presence in CAW. Transcriptomic and metabolomic data were collected from these treated cells. The primary outcomes in this study, derived from these two untargeted molecular domains, are both the individual gene expression and metabolite abundance levels, as well as gene and metabolite signatures of co-expression and co-abundance. This design allowed us to assess the effects of, and nature of interactions between, the TT and CQA compounds, as well as interactions between these compounds and other unknown compounds in CAW, using these outcomes.

## 2. Results

### 2.1. Analysis of Culture Media Post-Incubation

Neuron culture media was collected and frozen after the 48 h treatment period at the time of cell harvest. The cultured media was then extracted with OstroPlate^®^ solid-phase extraction, and concentrations of selected phytochemicals were measured using liquid chromatography coupled to multiple reaction monitoring mass spectrometry (LC-MRM-MS; Appendix A). In many of the incubations, compound concentrations were somewhat higher than the pre-incubation time zero starting value, possibly due to evaporation of solvent during incubation (Appendix A). Interestingly, the post-incubation concentration of CQA metabolites was ten-fold lower in CAW incubations compared to incubations containing CQA alone or TTCQA. This suggests a rapid degradation of CQAs in the presence of other CAW constituents. CQA metabolites are known to undergo chemical changes in non-acidic solutions [35,36]. Triterpene-glycoside content (MS and AS) was similar in all cultures containing these compounds. Triterpene aglycones (MA and AA) were present in higher concentrations post-incubation in CAW than the TT or TTCQA cultures, potentially arising from precursors present in CAW, as they were considerably higher than the time 0 values (Appendix A). Differences in response to CAW compared to the compounds may in part be due to these concentration differences.

### 2.2. CAW and Its Constituent Compounds Induce Extensive Gene Expression Changes

Of the 40, 39 culture samples were of sufficient quality for RNA-seq processing, displaying a high level of inter-neuronal networking and neuro-spheres (n = 8 per treatment except for TTCQA, which had n = 7). RNA extracted from the samples was of high quality with an average RNA Integrity Number (RIN) of 9.9. For the metabolomics processing, we had 50 cell culture samples of sufficient quality (n = 10 per treatment).

To investigate both differentially expressed genes (DEG) and differentially abundant metabolites (DAM), we compared each of the four treatment groups to the vehicle control (CAW vs. Ctrl, TT vs. Ctrl, CQA vs. Ctrl, and TTCQA vs. Ctrl). RNA-seq genes with expression less than 10 counts in any of the samples were removed, leaving 18,491 genes for analysis. Our in-house library provided 192 metabolites for analysis. A false discovery rate cutoff of 0.05 was used for both DEG and DAM analyses. While there was a high degree of treatment-related gene expression change in the neurons (CAW 946 upregulated, 1721 downregulated; CQA 33 upregulated, 165 downregulated; TT 514 upregulated, 1.246 downregulated; TTCQA 303 upregulated, 678 downregulated), based on fold change (Appendix A), the same was not true for changes in metabolites. In contrast, the metabolomics analysis found only 3 downregulated metabolites for CAW, 8 upregulated and 7 downregulated metabolites for CQA, 3 upregulated and 5 downregulated metabolites for TT, and 12 upregulated and 8 downregulated metabolites for TTCQA (Appendix A). This difference in the magnitude of DEGs as compared to DAMs was largely because the full transcriptome (18,491) was evaluated, but only a select number of metabolites (192) was assessed. When compared by the percentage of affected genes or metabolites, relative to the total number tested, the difference is not as dramatic (transcriptome: 21.8% of the 18,491 genes had a change in expression; metabolome: 16.1% of the 192 metabolites had a change in abundance).

A principal component analysis (PCA) was used to assess the effect of each treatment relative to the control and also to compare the treatments to each other, using both the transcriptomic and metabolomic data. This work was performed with the PCA tools R package [37]. Metabolites and genes in the lowest decile of variability across samples were removed. There was not a strong separation of samples seen between treatments for both data types, as would be expected since some of these treatments contain the same compounds (Figure 1A). However, for the independent treatment vs. control analyses, there is greater separation of samples as the number of differentially expressed metabolites between the groups being compared increases (Figure 1C). The same is generally true with the transcriptomic data (Figure 1B). This confirmed that the treatments were having some effect.

### 2.3. Complex Interactions Were Seen Between CAW Compounds, Both Activating and Inactivating Genes

Transcriptomic analysis revealed many interactions were apparent between constituent compounds in CAW. Only 17% of the DEGs that are seen in the TT and CQA separate treatments are also seen in the combined TTCQA treatment. (Appendix A), However, there are 420 new DEGs seen in this combination, of which only 263 are seen in the full CAW treatment, indicating the potential ‘silencing’ of 157 DEGs by interactions with unknown additional compounds in CAW. Interestingly, 177 of the TT DEGs not seen in the TTCQA treatment return in the full CAW treatment, and there are only 26 DEGs in common to all four treatments **(**Appendix A). In the metabolomic data, only two metabolites downregulated by a constituent treatment (CQA) were seen in the full CAW treatment (Appendix A).

A more detailed classification of the different types of gene expression interactions (additive, synergistic, antagonistic) from the differential gene expression model between TT and CQA can be seen in Figure 2 (See Section 4.6).

For this analysis, a little less than half of the genes examined had a negative interaction between TT and CQA (n = 966). A substantial number of these genes had a significant effect from the TT treatment (n = 728), while very few had an effect from CQA (n = 33). There was an increased downregulation in the TTCQA treatment in 524 of these genes, 238 of which did not have a significant change from either TT or CQA (considered a negative interaction for the sake of this analysis). Other interactions reduced or neutralized significant upregulated effects seen in either TT or CQA (n = 443).

We saw a larger number of genes with a positive interaction between the TT and CQA treatments (n = 1272). Most of these were either downregulated by TT, CQA, or both and had their downregulation either diminished (n = 154) or neutralized (n = 888) in the TTCQA treatment. Interestingly, there were 182 genes that were not affected by either the TT or CQA treatment that were significantly upregulated by the TTCQA mixture.

Finally, there were 30 genes that did not have any interaction between the TT and CQA treatment, all of which were either upregulated (n = 21) or downregulated (n = 9) by only TT (Figure 2). A table of the parameter estimates for each treatment can be found in (Appendix A) for the TT and CQA gene expression interaction classification.

### 2.4. Multiple Modules Capturing a Diverse Variety of Compound Interactions Point to Distinct Cellular Mechanisms Affected by CAW and Its Constituent Compounds in the Transcriptomic Data

Weighted Correlation Gene Network Analysis (WCGNA) was used to find groups of genes expressed in a correlated manner across the 39 samples, irrespective of DEG status, indicating probable participation in a common biological function. The soft thresholding power was set at 12 to create a scale-free topology in the weighted co-expression network. The 18,491 genes clustered into 14 modules (see Section 4.7). A least squares linear method with post-hoc Tukey multiple comparison tests were used to identify treatment differences, relative to control, within these modules (FDR cutoff of 0.05). Eight of these modules showed significant eigengene differences between at least one of the treatments and the control. To functionally classify these modules, we performed an over-representation analysis using a hypergeometric test to see if the number of genes from a module mapped to a pathway was greater than would be expected by random chance. Genes in each module were used to enrich *Mus musculus* pathways from the Reactome Pathway Database using the ReactomePA package in R version 1.46.0 [38]. Genes in five of the eight modules significantly enriched molecular pathways in Reactome. Sample eigengene distributions for these five modules are shown in Figure 3 by treatment group.

The size of the five modules varied from 256 to 2490 genes. Module 1 (M1) was enriched for extracellular matrix organization and collagen biosynthesis pathways. CAW was the only treatment with a significant effect on this eigengene (70% upregulated DEGs) (Figure 3, Module 1). Fatty acid metabolism pathways were associated with Module 2 (M2), and three treatments show a significant effect (TT, TTCQA, and CAW; 80% downregulated DEGs) (Figure 3, Module 2). Module 3 (M3) enriched pathways included cellular response to stress and stimuli and were significantly impacted by the TT treatment (61% upregulated DEGs) (Figure 3, Module 3). Module 4 (M4) enriched pathways are primarily involved in immune system functions. Here again, only the TT treatment showed a significant impact (90% downregulated DEGs). (Figure 3, Module 4). Enriched pathways for Module 5 (M5) are mostly associated with Electron Transport and Mitochondrial Biogenesis, only affected by the CQA treatment (91% downregulated DEGs) (Figure 3, Module 5). The pathways mentioned here were the highest level. For a more detailed description of smaller enriched pathways see Appendix A.

Module 1 of Figure 3 also shows an associated heatmap of individual gene expression levels for this module with groups defined by hierarchical clustering. The two highest-level clusters show a reversal of expression patterns associated with the treatment of significant effect (CAW), with an upregulated cluster for the treatment group showing a mostly downregulated pattern in the control group, and vice versa. For heatmaps associated with Modules 2–5, see Appendix A. The number of module genes in the enriched pathways was generally small in our analysis, compared to the total number in the module (M1 = 6%, M2 = 2%, M3 = 11%, M4 = 21%, M5 = 4%). It is not unusual for the fraction of genes related to the primary biological function of the module to be less than 20% of all genes in that module. While not all genes in the module were associated with the enriched function(s), genes that were not in these pathways but had a strong relationship to the module could be (guilt by association) [39]. To evaluate the strength of the relationship of both enriched and non-enriched genes to the entire module, we assessed both module membership (MM) and intramodular connectivity (IC). MM, or kME, is the correlation between individual module gene expression and the eigenvalue of that module. IC, or kWithin, is the gene degree, or number of neighboring connections within the module, in the weighted gene co-expression network that we constructed. The MM of the genes in enriched pathways was evaluated using a *t*-test with Bonferroni correction and found that genes within four out of the five modules (M1, M2, M3, M4) had significantly higher MM scores than genes not found in enriched pathways. Interestingly, the MM of M5 was significantly lower than the non-enriched pathway MM in this module, having a weak negative correlation (r = −0.19 average) with the eigengene.

Next, the treatment DEGs contribution to each module was evaluated. While not all treatment DEGs were found in the five modules, the significant treatments, with respect to the eigengenes, had the highest number of significant DEGs in the module. CAW DEGs comprised 28% of the genes in M1, whereas all other treatment DEGs were less than 1% of this module. For M2, TT DEGs were 30% of the total module genes, TTCQA DEGs were 23%, CAW DEGs were 38%, and CQA DEGs were less than 1%. TT DEGs in M3 were 22% of total genes, and none of the other treatments were more than 10%. In M4, TT DEGs comprised 63% of the module, and none of the other treatment DEGs were more than 7% of the module. Treatment DEG contributions were lower in M5, but the significant eigengene treatment CQA still had the highest percentage of the module at 9%.

The MM measure was significantly higher for the DEGs associated with significant eigengene treatments compared to non-DEGs in the module in two modules, M2 and M4 (M2: TT 0.54 vs. 0.18, TTCQA 0.52 vs. 0.22, and CAW 0.58 vs. 0.12; M4: TT 0.68 vs. 0.31; Bonferroni adjusted *t*-test). No other significant treatment DEGs in the other three modules were found to have significantly higher MM measures.

Each module displays a distinctly different pattern of CAW compound interaction at the eigengene level (Figure 3, all modules). M1 (Extracellular Matrix Organization) shows an increasing effect as the number of compounds increases, reaching significance with the full CAW treatment. M2 (Fatty Acid Metabolism) shows the effect of the TT treatment carried through to the full CAW treatment. CQA does not have an effect in this module and does not seem to diminish the TT effect, nor do the additional unknown compounds in CAW. The significant effect of TT seen in M3 (Cellular Response to Stress) seems to be diminished by CQA and even further diminished by the unknown compounds. The same pattern is seen with the TT treatment in M4 (immune function), but here TT has a downregulating effect, whereas it had an upregulating effect in M3. In M5 (Electron Transport and Mitochondrial Biogenesis), only the CQA treatment had a significant effect, which seems to be negated through interactions with the TT compounds and not further impacted by interactions with unknown compounds in CAW.

Information about the individual genes in these five modules and the pathways they enrich can be found in Appendix A.

### 2.5. TT Treatment Has the Greatest Effect on Functions Associated with Metabolite Co-Abundance, and This Effect Is Diminished by Interactions with Other Compounds

A modified protocol for WCGNA, with normalized metabolite abundance values, was used to find groups of metabolites that were co-expressed across the 50 samples (10 from each treatment plus control) [40]. Using a soft thresholding power of 12 to construct the co-abundance network, the 192 metabolites clustered into 6 modules (see Section 4.7). Two of the six modules had significant differences between at least one of the treatments and control (FDR cutoff of 0.05). A Tukey multiple comparison test was employed to identify the specific treatment differences in each of these two modules (Figure 4A). Metabolites from both modules significantly enriched 31 Reactome *Mus musculus* pathways for Module 1 and 345 pathways for Module 2 (see Appendix A for pathway information), using a hypergeometric test.

The majority of metabolites in Module 1 (M1) were amino acids (Figure 4C, M1), which was reflected in several of the 31 enriched pathways (‘branched-chain amino acid catabolism’, ‘phenylalanine and tyrosine metabolism’). Most of the metabolites in Module 2 (M2) were either nucleotides or amino acids (Figure 4C, M2). This was also reflected in several of the 345 enriched pathways for this module (‘metabolism of amino acids and derivatives’, ’metabolism of nucleotides’).

Information about individual metabolites in both modules can be found in Appendix A.

There were no enriched pathways in common between those associated with metabolite module M1 and the pathways associated with the five transcriptomic modules described in Section 2.4 However, there were common enriched pathways between metabolite module M2 and four of the transcriptomic modules (transcriptomic M1, M3, M4, M5). Some of these pathways include ‘signaling by PDGF’ for M1, ‘cellular responses to stress and stimuli’ for M3, ‘homeostasis’ for M4, and ‘respiratory electron transport’ for M5. For a complete list of the common pathways, see Appendix A.

### 2.6. Integration of Transcriptome and Metabolome Shows Little Overlap Between TT, CQA, and TTCQA in Significant Gene Activity, but Commonality in Pathways Affected

In the previous section, it was noted that there were several common biological pathways enriched with both genes and metabolites from the WGCNA modules generated for both omics domains. To focus more specifically on how an individual treatment impacts genes and metabolites that are functionally related to each other within a certain biological function (pathway), we performed an integration analysis as described in Section 4.8. This integration approach considered all DEGs for a specific treatment and then used a computational framework to find metabolites, from both the DAMs and WGCNA modules, that were also impacted by the same treatment and were statistically associated with the DEGs. This framework used network methods and prior knowledge of biological functions (pathways).

The 1760 DEGs and 133 DAMs (8 individual DAMs, 25 from Module 1, and 100 from Module 2) identified following TT treatment were used as seeds to construct the composite network as described in Section 4.8. This network was constructed without the necessity of additional connector genes/proteins to contain the maximum number of seeds possible. However, there were genes in the metabolite-gene network that were not differentially expressed by TT. There were 6 communities detected in the composite network that contained both a gene and metabolite seed, for a total of 37 metabolites and 42 genes in close functional relationship to each other (see Appendix A). Reactome pathway enrichment, conducted at the individual community level using both seed and non-seed genes, identified a total of 29 pathways containing at least one DEG for the TT treatment overall (see Appendix A). An example of the network derived from one TT community from the composite network is in Figure 5. Individual network figures for each community containing both seed metabolites and genes are in Appendix A.

The 198 DEGs and 115 DAMs (15 individual DAMs and 100 from Module 2) affected by CQA treatment were used as seeds. In this case, connector genes/proteins were used for maximum seed inclusion in the composite network, which yielded five communities, meeting our criteria of containing both a seed metabolite and a gene. These communities contained a total of 23 seed metabolites and 26 seed genes (see Appendix A), and these DEGs contributed to the enrichment of 74 pathways (see Appendix A). Module network figures are in Appendix A.

The 981 DEGs and 20 individual DAMs (no Module metabolites) altered with TTCQA treatment were used as seeds. Again, connector genes/proteins were used to construct the composite network, yielding two communities of interest, containing a total of 2 seed metabolites and 24 seed genes (see Appendix A), contributing to the enrichment of 90 pathways (see Appendix A). Module network figures are in Appendix A.

No composite network communities built using CAW seed genes and metabolites met the necessary criteria.

Next, pathway and gene overlap were evaluated between treatments using the composite network module level data, as described above. Comparisons were performed between TT and CQA, TTCQA and TT, and TTCQA and CQA. There were very few genes in common between any of these treatments (GPT for TT and CQA, CYP4F15 and ACOT5 for TTCQA and TT, and none for TTCQA and CQA). However, while there was only one pathway in common between TT and CQA, there were 6 pathways in common between TTCQA and TT and 34 between TTCQA and CQA.

### 2.7. Functional Relationships of DEGs for All Treatment Groups Are Not by Random Chance

A network approach was used to determine the overall functional similarity of the genes differentially expressed by each treatment. First, a protein–protein interaction (PPI) network was constructed using the STRING Database [41]. Protein interactions with a physical interaction score ≥ 400 (scale 0–1000), which is a moderate to high confidence in the interaction, were kept, and then the DEGs from each treatment were mapped onto this network, and the average network shortest path between the DEGs within each treatment separately was calculated. Permutation testing (1000 permutations) in the PPI network was then conducted by randomly distributing the same number of genes that were differentially expressed in each treatment. This was performed to see if the average shortest path distance between the DEGs for a specific treatment was closer than would be expected by random distribution. DEGs associated with all four treatments were significantly closer in the STRING PPI than would be expected by chance, as was also the case with the DEGs found in the five co-expression modules.

## 3. Discussion

Herbal or botanical healthcare products generally consist of complex mixtures in the form of powders or extracts of single or multiple herbs. The effects of the product may be due to the bioactivities of individual compounds present in the mixture but may also result from complex interactions between the different components. Studying interactions that may occur between multiple components of a botanical mixture presents a significant challenge. Several methods exist to understand these interactions, including the use of isobolograms of bioactivity vs. the proportion of the components being tested and biochemometric approaches where the activity of mixtures including or missing specific components is evaluated [16,42,43,44,45,46,47]. Untargeted transcriptomic analysis has previously been used to study individual and combined effects of herbs and phytochemicals in neuroglial cell lines [14,15].

In the present study, the primary outcomes derived from two untargeted omics domains, as well as their integration, were used to evaluate interactions between components of *Centella asiatica* water extract, CAW, which contains a complex set of components as previously reported [31,48]. This study compared different subsets of CAW constituent compounds, as well as the complete CAW extract, in a nested compound design.

While the earlier studies used targeted analyses of specific effects of the extract, here the effects of the nested subsets of compounds were compared in an untargeted fashion to identify broad patterns of activity and interactions for further research. For example, previous in vitro synergy experiments have generally evaluated a single biological activity in a targeted fashion, such as cell viability or an anti-microbial effect, often in a pairwise fashion between two compounds [44], and previous systems biology in silico methods are reliant on computational predictions [43]. To our knowledge, this is the first in vitro approach using systems biology methods to study compound interaction effects on bioactivity in a complex plant extract with a nested compound design with two untargeted molecular domains.

At the individual gene level, the TT treatment stood out in the gene expression results, with the most activated genes of the three select compound treatments (TT, CQA, and TTCQA) and the most activated genes that were still retained in the full CAW treatment, as compared to those retained from the CQA and TTCQA treatments in CAW. This would be consistent with the body of previous research focusing on TTs as the main bioactive compound in *Centella asiatica,* with activity demonstrated for many health conditions, including neurodegenerative, dermatological, and others [17]. However, in this study, there were many complex interactions reflected in the overlap of activated genes between TTs and other treatments. A large portion of gene activity seen with either the TT or CQA treatments was not seen when the two groups were combined or in the complete CAW extract, with 66% of the TT and CQA gene effects disappearing with TTCQA and 56% of these effects disappearing with CAW. Interestingly, 10% of gene expression lost on combining compounds in TTCQA is restored with the full CAW treatment, which suggests that the additional compounds within CAW are modulating these effects. Based on the post-incubation cultured media analysis, this return of gene activity in CAW could in part be modulated by the degradation of CQA compounds, which appear to inhibit the TT activity in the TTCQA treatment. Since there was no apparent degradation of either TT or CQA compounds seen in the TTCQA treatment (the post-incubation concentrations remained relatively unchanged for CQA in TTCQA), it could be assumed that the TT inactivation was through some other type of compound inhibition. There was also a large amount of new gene activity unique to TTCQA and CAW that was not observed in TT or CQA groups alone, suggesting the presence of interactions between TT and CQA (in the case of TTCQA) and/or the involvement of additional compounds in the case of CAW for interactions with these genes. Further studies are needed to identify whether DEGs unique to CAW could be the result of interactions between TT or CQA and other compounds within the extract.

Continuing with individual genes, the combination TTCQA treatment inactivated many genes whose expression was altered by TT. Most of this interaction is positive on genes downregulated by TT, with no effect by CQA alone. However, there is also a large number of upregulated TT genes neutralized by negative interaction with CQA (Figure 2 and Appendix A).

Some of these inactivated genes explain the therapeutic mechanisms of TT. The effect of TT on these genes not only has a therapeutic effect on neurodegenerative diseases but also on endocrine, dermatological, cardiovascular, digestive, respiratory, gynecological, and rheumatoid diseases (Asiatic acid impact on *AKT*, *mTOR*, *NF-κB*, *BDNF*, *CPT-1*, *SOX2*, *BCL2*, *IL18*, *CASP-3*, and *NLRP3* modulates the therapeutic effect for some neurological, digestive, dermatological, endocrine, cardiovascular, and respiratory conditions. Asiaticoside’s impact on *NF-κB*, *MAPK*, *BCL2*, *TLR4*, *TRAF6*, *IL18*, and *IL10* modulates the therapeutic effect for some neurological, endocrine, cardiovascular, digestive, and respiratory conditions. Additionally, madecassoside impacts *MAPK*, *TLR2*, and *IL10*, modulating the therapeutic effect for some neurological, endocrine, dermatological, and rheumatoid conditions [17,49]. This finding could suggest a therapeutic rationale for not combining TT and CQA compounds, i.e., using isolated TT compounds. While TT compounds are more widely studied, recent research has also shown health benefits from the CQA compounds, such as ameliorating cognitive impairment in an Alzheimer’s mouse model [28].

When evaluating higher-level functions using gene co-expression modules affected by each treatment, TT again showed the greatest effect across both the transcriptomics modules and metabolomics modules. However, there were very different patterns of interaction with TT and other compounds for these different modules. In Module 2, which was enriched for fatty acid metabolism and had a dominant gene downregulation effect from TT, there was virtually no interaction with CQA or unknown compounds. This module effect was still seen with TTCQA and CAW. One of the pathways within fatty acid metabolism was arachidonic acid metabolism. Downregulation of this pathway could explain some of the anti-inflammatory and immune-modulating effects of both TT and CAW [17]. With both Module 3 and Module 4, the significant effect of TT, seen with eigengenes (Figure 3), appears to be diminished by CQA and then further diminished by unknown compounds in CAW. Both of the overall functions associated with these modules through the enrichment analysis (cellular response to stress and stimuli, immune functions) were previously researched for CA. While the pattern of compound interaction is the same in these two modules, the effect of that interaction is different. For Module 3 (cellular response to stress and stimuli), the effect is overall an upregulation of genes, which seems to be neutralized by interactions. The reverse is true for Module 4 (immune functions). However, both triterpene compounds and extracts of CA, water, or ethanol have shown effects in these functions in previous research [17]. However, most of these were in vivo studies. The metabolomic Module 1 had the same pattern of interaction and differential abundance as the transcriptomic Module 4, but the metabolomic Module 2 had a pattern of interaction distinct from any transcriptomic module. For this module, both TT and CQA have relatively equivalent significant effects that are apparently neutralized through their interaction and then further reduced by interactions with unknown compounds.

The significant effect for CAW seen in transcriptomic Module 1 (extracellular matrix organization) appears to be mediated through interactions of all the compounds tested in this experiment since no significant effect is seen until all compounds are together in the full CAW extract, with possibly a strong contribution from unknown compounds (Figure 3, Module 1). Collagen formation sub-pathways, within extracellular matrix organization, are dominant in this module. Interestingly, TT has been found to reduce the deposition of the extracellular matrix and is thus useful in treating liver, pulmonary, and other fibrotic conditions [50,51]. However, a methanol extract of *Centella asiatica* has been found to stimulate collagen and extracellular matrix formation, which is beneficial in certain dermatological conditions, as well as wound healing [17,49,52]. TT compounds alone have shown benefit in treating dermatological conditions, but this effect appears to be modulated through other mechanisms besides collagen formation [17].

The integration analysis conducted for this experiment was metabolomics-centric since we had a much smaller number of metabolites of interest than genes. Finding genes associated with these metabolites helped to focus on specific functions for the compound interaction analysis. This is likely why some of the functionality seen with the gene co-expression analysis is not reflected in this integration. Generally, functions seen here are more closely reflected in the metabolite co-abundance findings, except for fatty acid metabolism.

Distinctly different functions between the TT and CQA treatments were apparent at the pathway level based on this integration analysis. Some of the top pathways associated with DEGs in the TT integration included the metabolism of amino acids, nucleotides, and fatty acids. The top pathways associated with DEGs in the CQA integration included chromatin organization, epigenetic regulation of gene expression, and DNA repair. There were also functional variations across network community groups within a treatment (see Appendix A). Many of the same pathways seen with CQA are also seen in TTCQA, such as chromatin organization, epigenetic regulation of gene expression, and DNA repair. Interestingly, there is almost no overlap between these three treatments at the DEG level, but a substantial amount of overlap at the pathway level, particularly between TTCQA and CQA, suggesting that some functions are retained in the TTCQA combination but mediated through different genes.

The PPI network analysis is another ‘guilt by association’ approach that considers two genes in a PPI to have a more likely functional relationship the shorter the path between them [53]. There is also some evidence that the network proximity of DEGs in a PPI is associated with the therapeutic synergy of the associated compounds [54]. This analysis revealed that none of the collections of altered genes were distributed by random chance. This would be expected for collections of compounds with structural similarity, such as the four compounds in the TT treatment of the eight compounds in the CQA treatment. However, it is more surprising to see this phenomenon maintained within the combination of these two sets of compounds (TTCQA), especially in the large number of structurally diverse compounds seen with CAW. It is possible that the lack of random distribution seen in CAW is mediated through compound interactions, and even possibly driven by the two therapeutic classes of compounds.

Future analyses could address some of the limitations of this current study. This study sample size was relatively small and may not be able to detect smaller significant effects, particularly with the metabolomics differential analyses. Additionally, the individual experiments were not able to collect sufficient cells from the cultures to perform the two omics analyses on the same batch of neuronal cultures. This particularly impacts our methodological approach to omics integration. Using a knowledge-based approach has limitations due to incomplete annotation of biological functions. The two independent co-expression/abundant analyses were data-driven and likely a more complete picture of functional molecular relationships, although still subject to limitations from the use of known pathway data for the enrichment analysis. Even the co-expression/abundance analyses are based on the assumption of a linear relationship between biological molecules, which is not always the case. Some sample variability can be seen in the eigengenes and eigenmetabolites in Figure 3 and Figure 4A. While the two omics experiments were performed in separate batches, the samples within each omics domain were from the same culture batch, so this would not have a source of variation. This could be related to interactions at the cellular transport level.

Since this was an in vitro experiment, some of the findings may not translate at an organism level. Some compounds present in these treatments may not be absorbed when given orally, and, in general, most pharmacokinetic mechanisms are not accounted for in this type of experiment. Future studies should explore the in vivo absorption of these compounds and also examine metabolomic and transcriptomic changes in the brains of mice treated with TT, CQA, TTCQA, and CAW. In humans, the concentration of TT and CQA compounds and endogenous metabolites could be measured in plasma following treatment with these mixtures, although these may not correlate directly to levels in the brain or other sites of action of these compounds.

It is difficult to know how the concentrations used in this experiment relate to the medicinal use of *Centella asiatica* as both traditional and commercial preparations vary widely in composition and dosage and corresponding plasma and tissue concentrations of TT and CQA are not usually known. However, in a clinical trial, a *Centella asiatica* TT preparation that improved symptoms of diabetic neuropathy resulted in steady state total TT plasma levels of about 0.6 µM [55]. Human pharmacokinetic studies of 2 g and 4 g doses of CAW [56] resulted in plasma C_max_ values for total TT of 0.5 and 1.2 µM, respectively. CQA levels resulting from *Centella asiatica* administration are more difficult to evaluate as they are widely distributed among commonly consumed food plants [57]. Nevertheless, the TT plasma concentrations seen in humans were of a similar order to the in vitro concentrations used in this study, supporting the potential clinical relevance of the data obtained here.

In conclusion, this study revealed some compelling patterns for further investigation. Four gene expression modules and two metabolite modules were altered by the 4 types of treatments applied. This methodology demonstrated the existence of both negative and positive interactions between TT, CQA, and additional compounds found in CAW on the transcriptome and metabolome of mouse primary cortical neurons. It will be interesting to confirm the pathways that emerged from this study in targeted future experiments. Additionally, it will be informative to evaluate how the phenotypic endpoints that previous work has shown to be affected by CAW are impacted by TT, CQA, and TTCQA treatment.

## 4. Materials and Methods

### 4.1. Mouse Primary Cortical Neuron Cell Cultures

Mouse primary cortical neurons were isolated and cultured (37 °C) following a previously published protocol [58]. Briefly, six-well plates were coated with poly-l-lysine (PLL; Sigma, Burlington, MA, USA), three days prior to primary neuron harvest. After one day, the PLL was removed, and the plates were washed with double distilled water. Following the wash, plating media (Minimal Essential Media, Fetal Bovine Serum 4.6%, Glucose 0.55%, Antibiotic-antimycotic: 10,000 U/mL Penicillin 10,000 U/mL Streptomycin and 25 µg of amphotericin B/mL, (Invitrogen, Carlsbad, CA, USA)) was added to each well for the remaining two days. Then, neurons were isolated from the cortices of C57BL6 mouse embryos at embryonic days 16–18, incubated in a mixture of HBSS and trypsin (2.5%) (Gibco, Grand Island, NY, USA) at 37 °C, then dissociated. Neurons were plated onto six-well plates (1 million/well) and incubated (37 °C) for 3 h after which the plating media was replaced with supplemented neurobasal media (with B27 1:50 dilution, Glutamax 1:100 dilution, antibiotic-antimycotic: penicillin 10,000 U/mL, streptomycin 10,000 U/mL and amphotericin B 25 μg/mL, (Invitrogen, Carlsbad, CA, USA)). Neurons were then cultured for five days at 37 °C before receiving experimental treatments.

### 4.2. Treatments

*Centella asiatica* dried plant material (aerial parts; batch number X20090016) was purchased from Oregon’s Wild Harvest (OWH; Redmond, OR, USA). Voucher samples were deposited in the BENFRA Center laboratories at Oregon Health & Science University (code number BEN-CA-6) and at the Herbarium at Oregon State University (code number OSC-V-265416). A water extract (CAW) was prepared by reflux extraction of the dried plant aerial parts minus the root, filtering, and lyophilization [23,24,25,59]. The ratio of dried plant material to dried extract was 5:1. The CAW treatment was prepared from this dried extract (0.050 mg/mL in 0.025% *v*/*v* aqueous methanol vehicle in each well). The triterpene (TT) and caffeoylquinic acid (CQA) content in the CAW was determined using liquid chromatography coupled with multiple reaction monitoring mass spectrometry (LC-MRM-MS) as previously described [31]. The triterpene (TT) treatment was prepared (0.025% *v*/*v* aqueous methanol vehicle final concentration in the cell culture) with madecassoside (1.7945 µg/mL; 1.84 µM), asiaticoside (0.7376 µg/mL; 0.769 µM), madecassic acid (0.0377 µg/mL; 0.073 µM), and asiatic acid (0.021 µg/mL; 0.043 µM) concentrations equivalent to that in the CAW 50 µg/mL solution (LC-MRM-MS verified concentrations). The caffeoylquinic acid (CQA) solution was prepared in the same fashion using chlorogenic acid (0.3749 µg/mL; 1.06 µM), neo-chlorogenic acid (0.1695 µg/mL; 0.478 µM), crypto-chlorogenic acid (0.1492 µg/mL; 0.421 µM), 3,4-dicaffeoylqinic acid (0.1146 µg/mL; 0.221 µM), 3,5-dicaffeoylqinic acid (0.0884 µg/mL; 0.171 µM), 4,5-dicaffeoylqinic acid (0.0975 µg/mL; 0.189 µM), 1,3-dicaffeoylqinic acid (0.1291 µg/mL; 0.250 µM), and 1,5-dicaffeoylqinic acid (0.1946 µg/mL; 0.377 µM), (LC-MRM-MS verified concentrations). The triterpene and caffeoylquinic acid (TTCQA) solution was prepared using the 12 compounds that were used in the TT and CQA treatments, prepared in the same fashion. The control vehicle consisted of aqueous methanol (0.025% *v*/*v*). These are the stock treatment ‘theoretical values’. These stock treatment solutions were then quantified using LC-MRM-MS as described in Section 4.3. These are the expected time zero values and can be found in the first four lines of Appendix A. Purified reference *Centella asiatica* TT and CQA were purchased from Chemfaces (Wuhan, Hubei, Peoples Republic of China).

CAW or compound treatments were initiated after five days of incubation and lasted for 48 h. There were four compound treatments (CAW, TT, CQA, TTCQA) and one vehicle control treatment. After the 48 h treatment period, an aliquot of medium from each well was collected for analysis of post-incubation cultured media compound concentrations, and cells were harvested by trypsinization (0.25% trypsin in Hanks’ balanced salt solution, 37 °C, 5 min) after media was removed and cells rinsed 1× with phosphate-buffered saline. Detached cells were then centrifuged at 300× *g* for 5 min at room temperature and the resulting cell pellet was flash-frozen and stored at −80 °C until analysis. Eight samples for each experimental condition were prepared for RNA sequencing and ten samples were prepared for each condition for metabolomics analyses.

### 4.3. Analysis of Compound Concentrations in the Media After 48 h of Treatment

For quantifying 12 CA phytochemical marker compounds (3 mono-CQA, 5 di-CQAs, and 4 TTs) in media, a SPE method was developed that removes phospholipids and proteins (using Ostro^®^ Protein Precipitation and Phospholipid Removal Plate (Waters Corp., Milford, MA, USA)) prior to LC-MRM-MS analysis. Each media sample was thawed and a 100 µL aliquot was transferred into an Eppendorf tube, that contained 300 µL methanol containing 0.1% formic acid and 5 µL digoxin-d3 (10 µg/mL) as an internal standard. Media sample was loaded into an Ostro plate well and 300 µL acetonitrile containing 0.1% formic acid was added and mixed by pipetting up and down 3 times. The plate was placed onto a vacuum manifold and the elution solvent was drawn into glass inserts of the collecting plate. The 1-mL glass inserts containing the eluate were placed into a 2-mL micro-centrifuge tube and dried in Speed Vac for approximately 1 h to obtain solid residues. For reconstitution, 50 µL of 70% methanol containing 0.1% formic acid was added to the glass inserts and resuspended by pipetting up and down 3 times. Samples were centrifuged at 13,000 rpm at 4 °C for 10 min, and each supernatant was transferred to an LC-MS vial and stored at −20 °C until LC-MRM-MS analysis using a Waters Xevo TQXS system connected to a Waters UPLC I-class. The MS method details have been described by us previously [31]. Chromatograms from LC-MRM-MS, targeted to detect only TT and CQA compounds, are provided in Appendix A. These show the presence of the required compounds in the stock solutions for TT, CQA, TTCQA, and CAW and that their concentrations are in line with those of the CAW extract. An untargeted LC-qTOF-MS chromatogram of all compounds detected in the CAW extract can be found in Appendix A showing the presence of additional compounds. Previous studies employing LC-qTOF-MS with a slower solvent gradient allowed the detection and annotation of 117 compounds in CAW extracts [48].

Cell pellets were resuspended in RLT-β-mercaptoethanol (QIAGEN) lysis buffer and quickly frozen for RNA extraction at the Oregon Health & Science University (OHSU) Gene Profiling Shared Resource. RNA quality was assessed at the GPSR and assigned an RNA Integrity Number (RIN). Only samples with RIN ≥ 8.0 were considered for sequencing, which occurred at the OHSU Massively Parallel Sequencing Shared Resource facility. This was performed on a HiSeq 2500 (Illumina, San Diego, CA, USA) and aligned to the mm38 mouse genome with the Star aligner [60]. Subsequent QA/QC was performed using the MultiQC package version 1.22 [61].

### 4.4. Metabolomics

Cells for control and treatments (approximately 1 million each) were thawed on ice and transferred to Precellys Lysing Kit bead blender tube with pre-supplied beads (2 mL Tissue Homogenizing Mixed Beads Kit (CKMix), Item No. 10409, Cayman Chemicals, Ann Arbor, MI, USA). Extraction solvents (500 μL, cold 80% methanol in water containing 12-[[(cyclohexylamino)carbonyl]amino]-dodecanoic acid (CUDA, 20 ppm, 10 μL)) were added to each vial, and homogenized. The homogenates were incubated for 1 h at −20 °C, then centrifuged at 10,000 rpm at 4 °C for 10 min. Each supernatant was transferred to a new vial and dried using a Speed Vac. Each residue was reconstituted in a cold 50% aqueous acetonitrile, vortexed for 20 s, and centrifuged at 10,000 rpm at 4 °C for 10 min. Each supernatant was transferred to an LC-MS vial and stored at −80 °C until LC-MS analysis. Quality control (QC) samples were generated by pooling 10-µL aliquots from each sample extract and were analyzed with samples. Blanks contained only extraction solvent.

The LC-MS/MS metabolomics workflow was described by us previously (see [62]) and entails two chromatographic methods: (1) Reverse-phase liquid chromatography was carried out using ACE Excel C18-PFP (1.7 μm, 2.1 mm × 100 mm). The sample injection volume was 5 μL, and the flow rate was 0.5 mL/min. The mobile phases consisted of water (A) and acetonitrile (B), both containing 0.1% formic acid. The gradient was as follows: initial hold at 5% B for 3 min, then increase to 80% B in 13 min, held until 16 min, then shift to 5% B and hold for equilibration from 16.5 min to 20.5 min. The column oven temperature was maintained at 30 °C. (2) Hydrophilic interaction liquid chromatography, the method was identical to that previously reported [62].

For both chromatographic methods ESI-MS/MS data were acquired in positive and negative ionization modes using a quadrupole time-of-flight MS system (AB Sciex tripleTOF 5600; Framingham, MA, USA). Individual metabolites were annotated using an existing in-house library based on retention time, exact mass, and MS/MS of 650 compounds compiled in the Mass Spectrometry Metabolite Library of Standards (MSMLS, IROA Technologies, Bolton, MA, USA). Raw data files were imported and processed using PeakView (ver. 1.2, Sciex) and MultiQuant (ver. 3.0.2, Sciex) software. Metabolites were verified using chromatographic retention time (error < 10%), accurate mass (error < 10 ppm), MS/MS fragmentation (score > 70), and isotopic pattern (error < 20%). The RP-C18 LC method yielded 117 metabolites (58 metabolites in positive ion mode and 59 in negative ion mode), and the HILIC method resulted in 234 metabolites (135 metabolites in positive ion mode and 119 in negative ion mode). The metabolites that were assigned in both methods were evaluated and the metabolites detected with the lower coefficient variation (CV) value in the QC samples were kept. This evaluation resulted in 192 high confident unique metabolites (178 metabolites (<20% QC CV), 10 metabolites (<30% QC CV), and 4 metabolites (>30% QC CV), Appendix A).

### 4.5. Differential Analyses for Gene Expression and Metabolite Abundance

For differential gene expression, the limma-voom package was used, a modification of the limma package designed for RNA-seq count data [63]. Genes were removed from the data if they did not have a transcript count of at least 10 in each sample. Next, a log_2_ transformation of counts per million (l cpm) was performed, followed by a trimmed mean of M-values (TMM) normalization. Heteroscedasticity was controlled with the voom function to meet the assumptions of a linear method. Differential calculations were conducted between each of the four treatments (TT, CQA, TTCQA, CAW) and the control (FDR cutoff 0.05).

For metabolite abundance, the MetaboAnalyst package version 5.0 was used [64]. The data were median normalized, log_2_ transformed, and then Pareto scaled (Appendix A). *t*-tests were performed to compare each treatment to the control for all 192 metabolites (FDR cutoff 0.05).

### 4.6. TT and CQA Gene Expression Interaction Classification

To assess the interaction between the TT and CQA treatments on normalized gene expression data, parameter estimates were derived from a least squares linear method with the control group as a reference, using the limma R package version 3.58.1 [63]. For each group, the effects (parameter estimates) of each treatment were assessed relative to the control. Next, the sum of the effects of TT and CQA treatments was contrasted with the effect of the TTCQA treatment using the contrasts.fit eBayes functions in limma. Any gene found to be differentially expressed by any of these three treatments was examined for interaction (N = 2268). We included genes that might not be differentially expressed by either one or two of the other treatments to see if an interaction might create or inhibit a significant expression.

If the TTCQA effect on the expression of a particular gene was greater than the sum of the TT and CQA effects, the interaction was classified as positive, or synergistic. If the TTCQA effect was less than the sum, the interaction was classified as negative, or antagonistic. If they were equal, the interaction was classified as an additive, or no interaction [65]. Since these coefficients represent log_2_ fold changes, negative values represent downregulation and positive values represent upregulation. So, any negative interaction that decreases this coefficient value would potentially increase downregulation, and positive interactions, as well as additive, would increase upregulation.

### 4.7. Weighted Gene Correlation Network Analysis (WGCNA Gene Co-Expression and Metabolite Co-Abundance)

The WGCNA package in R [66] was used for both RNA-seq and metabolomic data [40]. Both data types were normalized for the creation of this network by variance stabilizing transformation. To create a scale-free network, Pearson correlation was first used to construct an adjacency matrix for all gene or metabolite pairs using a soft thresholding power of 12 for both data types. The matrix was transformed into a topological overlap matrix (TOM) which was then converted to a dissimilarity matrix (1-TOM). Hierarchical clustering was applied to the final matrix to identify clusters of genes or metabolites with similar expression or abundance profiles. The minimum node size was 5 for metabolomics data and 30 for RNA-seq data. Module eigengenes and eigenmetabolites (first principal components) were calculated using the moduleEigengenes of WGCNA. Module membership for each gene or metabolite was calculated as the correlation between the normalized expression or abundance values with the module eigengene or eigenmetabolite and was calculated with the signedKME in WGCNA. Intramodular connectivity was the degree of each gene or metabolite in the adjacency matrix described above, calculated with the intramodular connectivity function in WGCNA.

Following the construction of these modules in each of the two omics domains, we then tested for differences between each treatment and control seen at the eigengene or eigenmetabolite level.

### 4.8. Integration of Transcriptomics and Metabolomics Data

A modification of a protocol previously used to integrate transcriptomic and metabolomic data for the study of neurodegenerative disease [67] was used to obtain a more comprehensive functional perspective for comparing the four treatments. This protocol was modified to fit the limitations of the data. Since this project was a repeated experiment design, with the transcriptome coming from different samples than the metabolome, rather than a split design with both omics coming from the same samples, a ‘data-driven’ approach was not appropriate. A ‘data-driven’ approach would correlate expression and abundance levels (or eigenvalues) across domains to create correlation-based networks for further analyses. Instead, a ‘knowledge-based’ approach was used, with existing molecular pathway information, to construct networks within and between individual genes and metabolites. These networks and subsequent analyses were conducted using OMICSNET software version 2.0 [68].

Genes and metabolites associated with significant treatment effects, relative to control, were used as seeds to construct the ‘knowledge-based’ networks in OMICSNET. Analyses were performed separately for each treatment. Any gene differentially expressed by a treatment was used as a seed. Any individual metabolite affected by treatment or member of a metabolite module whose eigenmetabolite was affected by treatment was also used as a seed. Due to the sparse number of individual metabolites affected, we also included co-expressed metabolites. Seed genes were used to construct a primary protein–protein interaction (PPI) network using the STRING database [41]. Genes and proteins are used interchangeably here. Connector proteins (non-seed) were added if necessary to create a more fully connected network. Seed metabolites were used to first create a primary metabolite-protein network using the KEGG pathway database [69]. A secondary PPI was created using the proteins in the primary metabolite-protein network using STRING, adding connector proteins if necessary. Finally, a composite network was created by linking the three independent network layers with common nodes (genes/proteins). To control the network size, the ‘Minimum network’ function was used in OMICSNET to create the smallest network possible which still contained as many seeds as possible.

Once an integrated network was created, tightly clustered communities of nodes were detected using the Walktrap algorithm [70]. Only communities that contained both seed metabolites and seed genes were further analyzed. These communities identified close functional relationships between genes and metabolites both affected by the same treatment. Subsequent Reactome pathway over-representation analysis was conducted using all genes in a community (seeds and connectors) using ReactomePA [38]. A comparison of pathways impacted by each treatment was then performed to identify overlapping and distinct functions using the integrated omics data (Figure 6).

## Figures and Tables

**Figure 1 pharmaceuticals-18-00019-f001:**
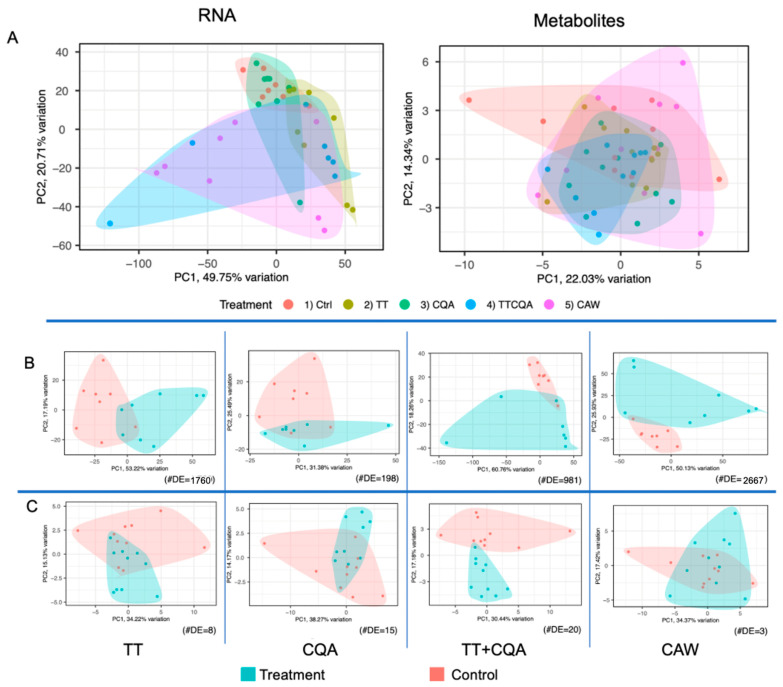
PCA plots for the two untargeted molecular domains of metabolomics and transcriptomics. (**A**) All treatments plotted together, including the vehicle control, for either metabolic or RNA-seq data. (**B**) RNA-seq data, each treatment compared individually to control. (**C**) Metabolic data, each treatment compared individually to control. For (**B**,**C**), the number of differentially expressed metabolites or genes (#DE) is shown in the lower right corner of each plot. The plots are also ordered left to right by increasing number of CA compounds in each treatment.

**Figure 2 pharmaceuticals-18-00019-f002:**
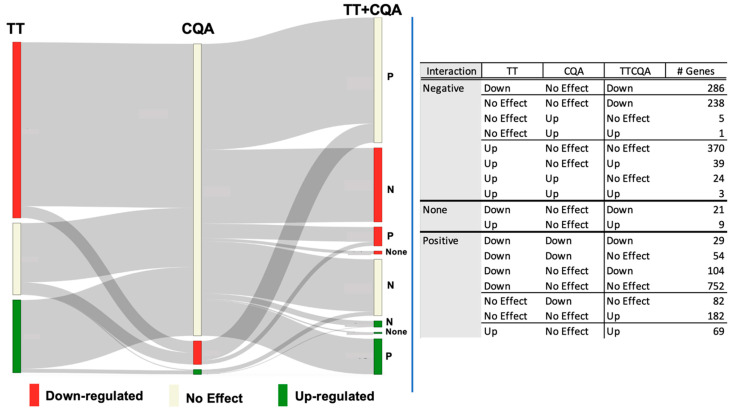
Effect on gene expression of the TT and CQA compounds administered separately and interactions between these groups observed in the TTCQA-treated samples. The figure on the left shows a mapping of the interactions listed in the table on the right. Any gene with significant expression changes, relative to control, seen with any of the TT, CQA, or TTCQA treatments are represented in the figure and table (N = 2268 unique genes). The relative number of genes and the expression status are represented within each of the three columns labeled TT, CQA, or TTCQA (red = significant downregulation, green = significant upregulation, beige = no significant expression changes seen with this treatment). The TT and CQA compound ‘Interaction effect’ for the combined group is shown next to the TTCQA column (‘P’ = positive, or synergistic; ‘N’ = negative, or antagonistic; ‘none’ = additive, or no interaction). The gray ribbons show the number of genes interacting between TT and CQA, the interaction status, and the final expression status in the TTCQA treatment. The table on the right shows the number of genes in each category. As an example, a positive interaction has a higher fold change value in the TTCQA treatment than the sum of the two-fold change values from TT and CQA, and a negative interaction has a lower TTCQA fold change than the sum of TT and CQA fold changes. So a positive interaction could still result in a downregulated or unregulated TTCQA gene, and a negative interaction could result in an upregulated or unregulated TTCQA gene.

**Figure 3 pharmaceuticals-18-00019-f003:**
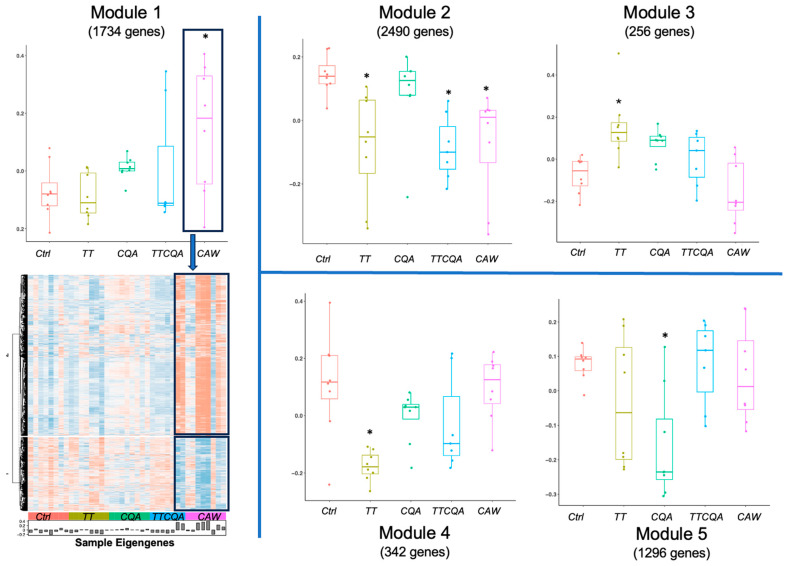
Sample eigengene distributions for the five gene co-expression modules with significant eigengene differences between at least one treatment and control. Boxplots for module eigengene distributions by each treatment group (* *p*-value ≤ 0.05). Module 1 = Extracellular Matrix Organization and Collagen Biosynthesis, Module 2 = Fatty Acid Metabolism, Module 3 = Cellular Response to Stress and Stimuli, Module 4 = Immune System, Module 5 = Electron Transport and Mitochondrial Biogenesis. Module 1 also shows a heat map of individual gene expression levels (blue = low, red = high). The highlighted section represents samples in the significant treatment for this module. See Appendix A for heatmaps for Modules 2–5.

**Figure 4 pharmaceuticals-18-00019-f004:**
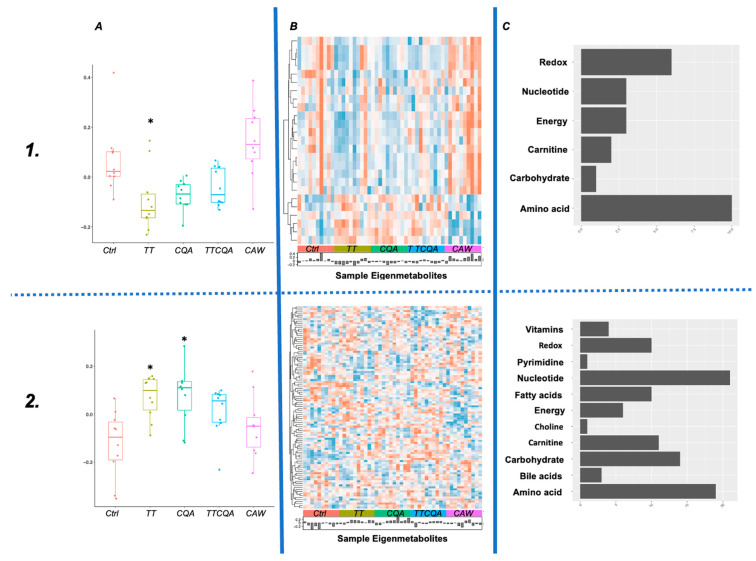
Metabolite co-abundance modules with significant eigenmetabolite differences between at least one treatment and control (**1.** Module 1 n = 25 metabolites, **2.** Module 2 n = 100 metabolites). (**A**) Boxplots for module eigenmetabolite distributions by each treatment group (* adj *p*-value ≤ 0.05). (**B**) Heatmap of individually scaled metabolite abundance for the module (red = high abundance, blue = low abundance). Samples are ordered by treatment group and labeled on the horizontal axis. Histogram shows eigenmetabolite values for each sample. (**C**) Metabolite category counts for all metabolites in the module.

**Figure 5 pharmaceuticals-18-00019-f005:**
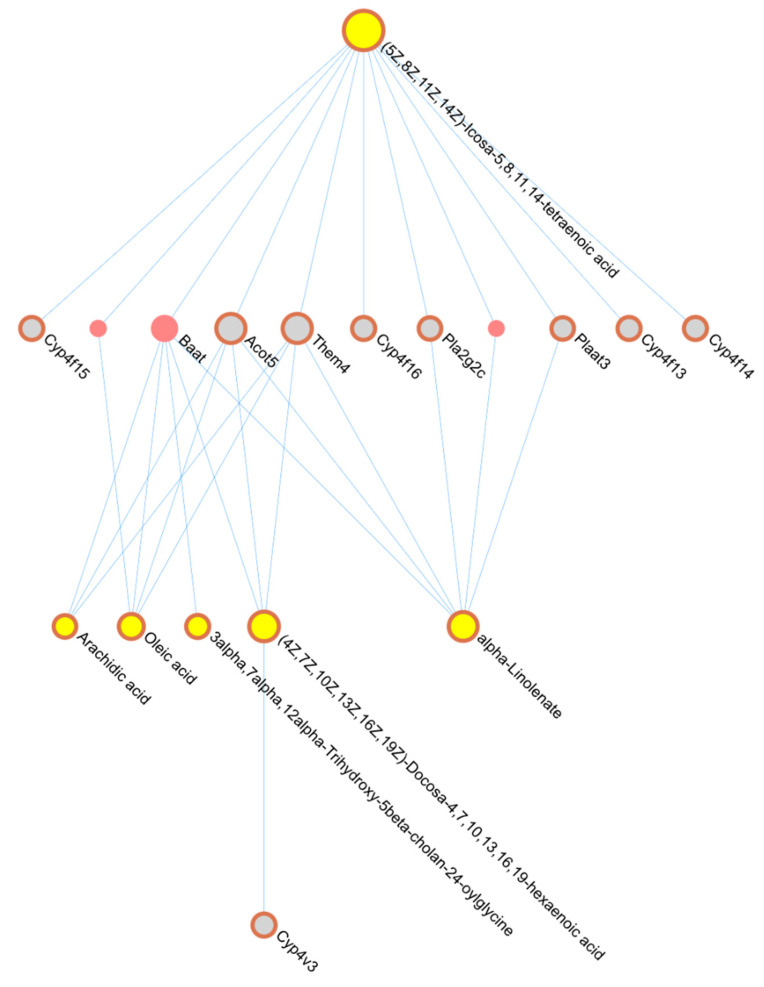
TT Integration Community Network Example. Network is derived from seed genes and metabolites in one community of the TT integrated composite network. Seed genes and metabolites are outlined in red. Genes are indicated in gray, and metabolites in yellow. Connector genes (not seed genes) are the red circles that are not outlined. The size of the circles is related to the number of network connections, or degree. This network enriched mostly fatty acid metabolism pathways.

**Figure 6 pharmaceuticals-18-00019-f006:**
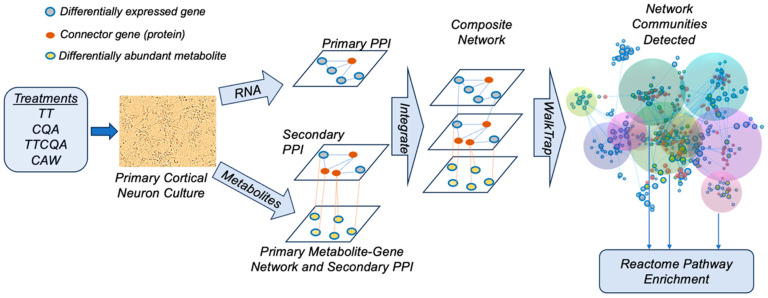
Omics integration methodology. Individual seed networks are shown in the center of the figure. For metabolite seeds two networks are shown, a primary metabolite-gene network and a secondary PPI created from the genes in the primary network. Seed genes are shown in this secondary PPI if present, but are not used to construct this network. For the seed genes, only one primary PPI network is created. The red connector (non-seed) genes (protein) are added to both PPIs, if necessary, to increase connectivity in the overall network for community detection that will better contain seeds of interest.

## Data Availability

The original contributions presented in this study are included in the article/Appendix A. Further inquiries can be directed to the corresponding author for RNAseq files.

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
