# Peer review of "Multi-Omics Analysis in Mouse Primary Cortical Neurons Reveals Complex Positive and Negative Biological Interactions Between Constituent Compounds of Centella asiatica"

_pharmaceuticals, 2024, doi:10.3390/ph18010019_

Round 1
Reviewer 1 Report
Comments and Suggestions for Authors
The manuscript addresses an important topic, investigating the complex interactions between triterpenes (TT), caffeoylquinic acids (CQA), and other compounds in Centella asiatica water extract (CAW) using a multi-omics approach. Although the study provided several interesting findings, several issues in the methodology section undermine its rigour and reproducibility:
- There are multiple typographical and formatting inconsistencies throughout the manuscript that require attention. For example, “poly l-lysine” should be corrected to “poly-L-lysine,” “25 μg of amphotericin B/ml” would be clearer as “amphotericin B 25 μg/mL,” and “aqueous methanol (.025% v/v)” should be revised to “0.025% v/v.” Additionally, the phrase “Cell pellets pellets were resuspended in RLT-ß-mercaptoethanol (QIAGEN)” contains a duplicated word (“pellets”). Such errors detract from the professionalism of the manuscript.
- The description of the trypsinization step is problematic. The authors state: “concentrations and cells were harvested by trypsinization (0.25% trypsin in media, 37°C, 5 min).” However, as the complete medium used earlier contains 4.6% FBS (Section 4.1), trypsin activity would be inhibited in such a medium. If “media” does not refer to the complete medium, this should be clarified explicitly.
- The centrifugation step, described as “Detached cells were then centrifuged,” lacks critical details. The speed (g-force) and duration of centrifugation must be specified to ensure the reproducibility of the procedure.
- The reference “Choi et al., J Am Heart Assoc. 2019;8:e012809” is cited in the text but is missing from the reference list. This omission must be corrected.
- There is an unresolved placeholder in the following sentence: “(178 metabolites (<20%), 10 metabolites (<30%), and 4 metabolites (>30%) in QC CV values, Supp Fig SF.xxxx).” This oversight indicates incomplete editing and should be addressed before publication.
Overall, addressing these issues is essential to enhance the manuscript’s clarity and ensure the reproducibility of the methodology.
Author Response
Comment 1: There are multiple typographical and formatting inconsistencies throughout the manuscript that require attention. For example, “poly l-lysine” should be corrected to “poly-L-lysine,” “25 μg of amphotericin B/ml” would be clearer as “amphotericin B 25 μg/mL,” and “aqueous methanol (.025% v/v)” should be revised to “0.025% v/v.” Additionally, the phrase “Cell pellets pellets were resuspended in RLT-ß-mercaptoethanol (QIAGEN)” contains a duplicated word (“pellets”). Such errors detract from the professionalism of the manuscript .
Response 1: These have been corrected.
Comment 2: The description of the trypsinization step is problematic. The authors state: “concentrations and cells were harvested by trypsinization (0.25% trypsin in media, 37°C, 5 min).” However, as the complete medium used earlier contains 4.6% FBS (Section 4.1), trypsin activity would be inhibited in such a medium. If “media” does not refer to the complete medium, this should be clarified explicitly.
Response 2: Additional details of the trypsinization protocol have been added to Section 4.2.
Comment 3: The centrifugation step, described as “Detached cells were then centrifuged,” lacks critical details. The speed (g-force) and duration of centrifugation must be specified to ensure the reproducibility of the procedure.
Response 3: This information has been added to Section 4.2.
Comment 4: The reference “Choi et al., J Am Heart Assoc. 2019;8:e012809” is cited in the text but is missing from the reference list. This omission must be corrected.
Response 4: This has been added as reference 63.
Comment 5: There is an unresolved placeholder in the following sentence: “(178 metabolites (<20%), 10 metabolites (<30%), and 4 metabolites (>30%) in QC CV values, Supp Fig SF.xxxx).” This oversight indicates incomplete editing and should be addressed before publication.
Response 5: This has been addressed in Section 4.5, last sentence, highlighted in yellow. A new supplementary table has been added (ST13) to address this missing table reference.
Reviewer 2 Report
Comments and Suggestions for Authors
In this manuscript the authors investigate the effects of the aqueous extract of the plant Centella asiatica Urb on primary mouse neurons. Specifically, the authors treated cells with triterpenes, caffeoylquinic acids and aqueous extract of Centella asiatica, alone and in combination. Transcriptomic and metabolomic analyses were performed to investigate differentially expressed genes and differentially abundant metabolites, comparing each treatment with the vehicle control. These analyses highlighted the presence of interactions and differentially expressed genes with respect to the different treatments, with both negative and positive effects.
The study was well conducted and well described, but one important issue should be addressed before publication (manuscript acceptance).
The manuscript lacks data on the phenotypes of neuronal cell lines treated with compounds alone or in combination, as reported in the Introduction section (last paragraph at the end of page 2). In fact, the studies in the cited articles were performed on hippocampal neurons and not on cortical neurons. The authors should show the effect on cells (such as mitochondrial biogenesis and activation of endogenous antioxidant mechanisms) of the various compounds and their mixtures at the concentrations used in the study.
Minor points
Page 16 - Par. 4.3, line 25. The word "pellets" is doubled.
Page 17 - line 32. The number of the supplementary figure is missing.
Author Response
Comment 1: The manuscript lacks data on the phenotypes of neuronal cell lines treated with compounds alone or in combination, as reported in the Introduction section (last paragraph at the end of page 2). In fact, the studies in the cited articles were performed on hippocampal neurons and not on cortical neurons.
Response 1: While much of our published work in cultured primary neurons has focused on hippocampal neurons, we have reported that similar effects of CAW can be observed in vivo following oral CAW administration and the same phenotypic changes are seen in both the hippocampus and cortex suggesting that similar effects would be observed in primary neurons from both regions. Text has been added to the last paragraph of the introduction to clarify this point.
Comment 2: The authors should show the effect on cells (such as mitochondrial biogenesis and activation of endogenous antioxidant mechanisms) of the various compounds and their mixtures at the concentrations used in the study.
Response 2: We agree that this would be a useful study to perform. We are not able to conduct additional in vitro experiments to evaluate the effects of compound groups on those endpoints within the resubmission window, but we have added text to the final paragraph of the discussion noting that this is an important thing to confirm in future work.
Reviewer 3 Report
Comments and Suggestions for Authors
In this manuscript, Chamberlin and co-worker reports “Multi-omics analysis in mouse primary cortical neurons reveals complex positive and negative biological interactions between constituent compounds in Centella asiatica” This article describes how the multi-omics analysis is applied to reveal positive and negative biological interactions between consistuent compounds in Centella asiatica. Although the obtained results are preliminary ones, some interaction can be detected by this multi-omics analysis. This manuscript should be of interest to readers interdisciplinary areas of medicinal chemistry. In summary, I think that this manuscript is appropriate for publication in Pharmaceuticals.
Author Response
No comments were given that required a response.
Round 2
Reviewer 2 Report
Comments and Suggestions for Authors
All my concerns have been addressed exhaustively by the authors.